# Immunomodulatory and Anti-inflammatory Effects of Asiatic Acid in a DNCB-Induced Atopic Dermatitis Animal Model

**DOI:** 10.3390/nu13072448

**Published:** 2021-07-17

**Authors:** Gyo-Ha Moon, Yonghyeon Lee, Eun-Kyung Kim, Kang-Hyun Chung, Kwon-Jai Lee, Jeung-Hee An

**Affiliations:** 1Department of Food and Nutrition, KC University, Seoul 07661, Korea; moongyoha@naver.com (G.-H.M.); yh_0904@naver.com (Y.L.); 2Department of Food Science and Technology, Seoul National University of Science & Technology, Seoul 01811, Korea; carl@seoultech.ac.kr; 3Department of Food Science and Nutrition, Dong-A University, Busan 49315, Korea; ekkimkr@dau.ac.kr; 4Center for Silver-targeted Biomaterials, Brain Busan 21 Plus program, Dong-A University, Busan 49315, Korea; 5Department of H-LAC, Daejeon University, Daejeon 34520, Korea; kjlee@dju.kr

**Keywords:** asiatic acid, atopic dermatitis, anti-inflammatory, immunomodulatory

## Abstract

We examined the immunomodulatory and anti-inflammatory effects of asiatic acid (AA) in atopic dermatitis (AD). AA treatment (5–20 µg/mL) dose-dependently suppressed the tumor necrosis factor (TNF)-α level and interleukin (IL)-6 protein expression in interferon (IFN)-γ + TNF-α-treated HaCaT cells. The 2,4-dinitrocholrlbenzene (DNCB)-induced AD animal model was developed by administering two AA concentrations (30 and 75 mg/kg/d: AD + AA-L and AD + AA-H groups, respectively) for 18 days. Interestingly, AA treatment decreased AD skin lesions formation and affected other AD characteristics, such as increased ear thickness, lymph node and spleen size, dermal and epidermal thickness, collagen deposition, and mast cell infiltration in dorsal skin. In addition, in the DNCB-induced AD animal model, AA treatment downregulated the mRNA expression level of AD-related cytokines, such as Th1- (TNF-α and IL-1β and -12) and Th2 (IL-4, -5, -6, -13, and -31)-related cytokines as well as that of cyclooxygenase-2 and CXCL9. Moreover, in the AA treatment group, the protein level of inflammatory cytokines, including COX-2, IL-6, TNF-α, and IL-8, as well as the NF-κB and MAPK signaling pathways, were decreased. Overall, our study confirmed that AA administration inhibited AD skin lesion formation via enhancing immunomodulation and inhibiting inflammation. Thus, AA can be used as palliative medication for regulating AD symptoms.

## 1. Introduction

Atopic dermatitis (AD) is a chronic inflammatory skin disorder with several symptoms, such as pruritus, erythema, swelling, and cracked skin [1]. T helper 1 (Th1) and Th2 cells are involved in eliciting a strong type 2 immune response, which plays an important role in AD; the immune response includes a significant increase in mast cell infiltration and levels of Th2-related cytokine and serum immunoglobulin (Ig) E [2]. The predominant Th2-mediated response in acute AD can cause oozing and pruritic or erythematous papules [3]. The Th1 immune response in chronic AD causes lichenification or desquamation [3] and results in an immune response involving interferon (IFN)-γ, interleukin (IL)-1α and -2, and transforming growth factor (TGF)-β [3,4]. Th2-related cytokines, including IL-4, -5, and -13, increase the lesional and nonlesional skin surface area in acute AD [5]. In addition, Th2-related cytokines can express IFN-γ, which participates in a pro-inflammatory pathway by increasing the number of Th1 cells [2]. Furthermore, Th2 cells increase the amount of IgE through the proliferation of B cells [6]. Receptors on the surface of the mast cells bind to IgE and the mast cells can increase IL-6, -1β, and tumor necrosis factor (TNF)-α secretion [7].

Asiatic acid (AA) has been found in various plants, such *as Actinidia arguta, Maytenus procumbens, Psidium guajava, and Centella asiatica* [8]. AA is a pentacyclic triterpenoid that is substituted by a carboxy group at position 28 and hydroxy groups at positions 2, 3, and 23 (the 2α, 3β stereoisomer) [9]. AA improves lipid profiles, plasma levels of the oxidative stress markers TNF-α and NOx, and insulin sensitivity [10]. In addition, AA has anti-inflammatory properties as it decreases iNOS expression levels in high-fat-fed rats with metabolic syndrome [10]. AA inhibits the growth of MDA-MG-231 cells that inhibit breast cancer by reducing WAVE 3 and PI3K/protein kinase B (Akt) protein expression [11]. AA reduces the expression of anti-inflammatory cytokines, including iNOS, COX-2, and NF-κB, in an acetic acid-induced animal model [12]. In a carbon tetrachloride-induced liver damage rat model, AA was shown to have an anti-fibrotic effect as it reduces aspartate aminotransferase activity and alanine aminotransferase activity in the serum [13]. Some studies have reported that AA reduces lipopolysaccharide (LPS)-induced NF-κB activation and inflammatory mediator production via peroxisome proliferator-activated receptor-gamma in human gingival fibroblasts [14]. Thus, several studies have reported the anti-inflammatory [10,12], anti-breast cancer [11], anti-fibrotic [13], and anti-periodontitis [14] effects of AA. Nevertheless, to the best of our knowledge, no study has focused on the effects of AA on the immune and inflammatory mechanisms related to 2,4-dinitrocholrlbenzene (DNCB)-induced AD.

This study aimed to evaluate the efficacy of immunomodulatory and anti-inflammatory effects of AA in IFN-γ- and TNF-α-treated HaCaT cells and DNCB-induced AD BALB/c mouse models. Furthermore, this is the first study to confirm the immunomodulatory mechanisms of AA treatment in a DNCB-induced AD animal model. Our results indicate that AA treatment reduced the levels of Th1- and Th2-related cytokines and inflammatory cytokines as well as NF-κB and MAPK signaling expression. Furthermore, AA inhibited AD skin lesions formation and other AD characteristics, such as ear thickening, dermal and epidermal thickening, collagen deposition, and mast cell infiltration in DNCB-induced AD mice.

## 2. Materials and Methods

### 2.1. Cell Culture and Viability

AA was obtained from Sigma-Aldrich (St. Louis, MO, USA). Human keratinocyte (HaCaT) cells (ATCC; Manassas, VA, USA) were incubated at 37 °C under 5% CO_2_ in Dulbecco’s modified Eagle’s medium supplemented with 10% fetal bovine serum (Hyclone; Logan, UT, USA) and 1% penicillin-streptomycin antibiotic (GIBCO; Grand Island, NY, USA).

HaCaT cells (1 × 10^5^ cells/mL) were pretreated with AA (5, 10, and 20 μg/mL) for 1 h. Then, they were treated with 0.2 ng/mL IFN-γ and TNF-α for 24 h to induce inflammation. To determine the cell viability, the 3-(4,5-dimethylthiazol-2-yl)-2,5-diphenyltetazolium bromide assay (Promega, Madison, WI, USA) was used [15]. 

### 2.2. AA Administration in Model Animals

Eight-week-old female BALB/c mice were obtained from Samtako (Osan, Korea), housed in specific pathogen-free transparent plastic cages bedded with aspen chips in a special control room (12/12 h dark/light cycle, 40–45% relative humidity, and 20–21 °C), and provided a standard mouse diet and tap water ad libitum when untreated. The study was approved by the Institutional Animal Care and Use Committee of Konkuk University (KU19108) and performed following the relevant guidelines and regulations. BALB/c mice were randomly divided into the following six groups (*n* = 7 per group): the control group (CON group), AD-only group (0.25% DNCB sensitized in 200 µL 3:1 acetone/bean oil), AD + AA-L and AD + AA-H groups (30 and 75 mg/kg AA, respectively, in 100 µL sterilized drinking water), AA-H group (75 mg/kg AA, oral administration), and positive control AD + Pred group (5 mg/kg prednisolone (Youhanyanghaeng; Seoul, Korea) in 100 µL of sterilized drinking water) [16] (Figure 1). 

In the AD treatment groups, DNCB induction was repeatedly performed for local exposure of skin lesions [17]. Ear thickness was measured using a dial thickness gauge (Kori Seiki MFG Ltd.; Tokyo, Japan) after DNCB application for 18 d. The mice ears and dorsal skin were resected and histopathologically analyzed after 18 d. The lymph node and spleen weights were measured using the CP-224S electronic balance (Sartorius; Göttingen, Germany).

### 2.3. Histological Observation

The ear and dorsal skin tissue from DNCB-induced animal model was fixed using 10% formaldehyde for 24 h, and 4-µm-thick paraffin sections were stained with hematoxylin and eosin (H&E) for histopathological examination. Masson’s trichrome staining was performed to observe collagen deposition. The dorsal skin of mice was stained using toluidine blue. H&E-, Masson’s trichrome-, and toluidine blue-stained sections were observed under a light microscope at 100× magnification, and the images were analyzed using OptiView image analysis software (Korea Lab Tech; Seongnam-si, Korea).

### 2.4. Real-Time Polymerase Chain Reaction

The total RNA was extracted from the mouse dorsal tissue using TRIzol (Sigma-Aldrich, St. Louis, MO, USA), according to the manufacturer’s instructions [18]. Real-time PCR was performed in triplicate [15]. Target gene mRNA levels were normalized to GAPDH levels using the following formula:

Relative mRNA expression = 2^(ΔCt target gene − ΔCt GAPDH)^(1)
where Ct is the threshold cycle value. 

In each sample, the expression level of the analyzed gene was normalized to that of GAPDH and presented as a relative mRNA level. The primers used were: GAPDH: F (5′-CATGGCCTTCCGTGTTCCTA-3′), R (5′-TGTCATCATACTTGGCAGGTTTCT-3′); TNF-α: F (5′-AAGCCTGTAGCCCACGTCGTA-3′), R (5′-GGCACCACTAGTTGGTTGTCTTTG-3′); IL-1β: F (5′-ATAACCTGCTGGTGTGTGAC-3′), R (5′-AGGTGCTGATGTACCAGTTG-3′); IL-12: F (5′-TGCAACGTTGGAAAGGAAAG-3′), R (5′-TACCTACGCAGCCCTGATTG-3′); IL-4: F (5′-TCTCGAATGTACCAGGAGCCATATC-3′), R (5′-AGCACCTTGGAAGCCCTACAGA-3′); IL-5: F (5′-ACAGGAGAAGGGACGCCAT-3′), R (5′-GAAGCCGTACAGACGAGCTCA-3′); IL-6: F (5′-CCACTTCACAAGTCGGAGGCTTA-3′), R (5′-GCAAGTGCATCATCGTTGTTCATAC-3′); IL-10: F (5′-TCAGCTGTGTCTGGGCCACT-3′), R (5′-TTATGAGTAGGGACAGGAAGCCTCA-3′); IL-13: F(5′-GCAACATCAACAGGACCAGA-3′), R (5′-GTCAGGGAATCCAGGGCTAC-3′); IL-31: F (5′-TCGGTCATCATAGCACATCTGGAG-3′), R (5′-GCACAGTCCCTTTGGAGTTAAGTC-3′); IL-17: F (5′-TCCCCTCTGTCATCTGGGAAG-3′), R (5′-CTCGACCCTGAAAGTGAAGG-3′); COX-2: F (5′-GCCAGGCTGAACTTCGAAACA-3′), R (5′-GCTCACGAGGCCACTGATACCTA-3′); CXCL9: F (5′-GGAACCCTAGTGATAAGGAATGCA-3′), R (5′-TGTCATCATACTTGGCAGGTTTCT-3′).

### 2.5. Western Blotting

The cells and mouse dorsal tissues were homogenized using lysine buffers containing protease inhibitors (Roche; Mannheim, Germany). The total soluble protein content was assessed using the Bio-Rad protein kit (Bio-Rad Laboratories; Hercules, CA, USA). The proteins were electrophoresed and transferred to Immobilon-P transfer membranes (Millipore; Burlington, MA, USA). The membranes were blocked with 5% bovine serum albumin and incubated at 4 °C for 24 h with specific primary antibodies against phosphorylated-p38 (p-p38), *p*-protein kinase B (Akt), *p*-n N-terminal kinase (JNK), *p*-extracellular signal-regulated kinase 1/2 (ERK1/2), β-actin (Cell Signaling Technology; Beverly, MA, USA), TNF-α, NF-kB, iNOS, COX-2 (Abcam, Cambridge, MA, USA), IL-6, and IL-8 (Santa Cruz Biotechnology; Santa Cruz, CA, USA). The membranes were then incubated at 4 °C with goat anti-rabbit IgG (H + L) and a horseradish peroxidase-conjugated secondary antibody (Abcam). The protein bands were visualized using enhanced chemiluminescence, and densitometric analysis of the protein bands was performed using the C-DiGit Blot Scanner (Li-COR; Lincoln, NE, USA) and ImageJ software (NIH; Rockville, MD, USA). All data were normalized to the β-actin values.

### 2.6. Statistical Analyses

All statistical analyses were performed using SPSS version 18.0 (IBM; Chicago, IL, USA). Comparisons between experimental groups were performed using one-way analysis of variance with Duncan’s post-hoc tests. Data for each test are presented as the mean ± standard deviation. Statistical significance was set at *p* < 0.05.

## 3. Results

### 3.1. AA Decreased the Viability of TNF-α + IFN-γ-Treated HaCaT Cells

We studied the viability of 10–70 µg/mL AA-treated HaCaT cells to determine AA cytotoxicity (Figure 2A). Our result showed that treatment with AA at a concentration of ≤20 µg/mL was not cytotoxic. However, treatment with AA at a concentration of ≥30 µg/mL was cytotoxic as it decreased cell viability by 62.2%. Hence, we treated HaCaT cells with 5, 10, and 20 µg/mL AA for western blotting.

IL-6 and TNF-α protein expression levels were dose-dependently downregulated in TNF-α + IFN-γ-treated HaCaT cells after AA treatment, indicating its anti-inflammatory properties (Figure 2B). The IL-6 protein expression level in the TNF-α + IFN-γ-only group (75%) increased compared to that in the normal group. However, this elevated expression was ameliorated by 18%, 61%, and 79% after 5, 10, and 20 µg/mL AA treatment, respectively, which was higher than that in the TNF-α + IFN-γ only group. Furthermore, the TNF-α level was 76% higher in the TNF-α + IFN-γ only group than in the normal group (23%). The increased expression level of TNF-α was reduced by 30%, 52%, and 52% after 5, 10, and 20 µg/mL AA treatment, respectively. Thus, AA treatment had a concentration-dependent effect on the inflammation of TNF-α + IFN-γ-treated HaCaT cells.

### 3.2. AA Reduced AD Skin Lesions Formation and Ear Thickness

We applied DNCB on the earlobes and dorsal skin to determine the effects of oral AA administration on AD (Figure 3). DNCB administration resulted in the development of AD skin lesions, indicated by swelling, excoriation, edema, scarring, hemorrhage, and scaling in the ear. The ear thickness was measured after DNCB application for 18 d. Figure 3A shows that swelling, scarring, and infection in the AD-only group were more substantial than those in the CON group. Ear thickness in the AD-only group (0.445 ± 0.009 mm) was 2.4-fold greater than that in the CON group (0.214 ± 0.036 mm). Furthermore, ear thickness in the AA-treated groups (AD + AA-L and AD + AA-H) was lesser (1.1- and 1.2-fold, respectively) than that in the AD-only group (AD + AA-L: 0.403 ± 0.019 mm; AD + AA-H: 0.382 ± 0.010 mm). Interestingly, the ear thickness was also not significantly different (*p* > 0.05) between the AA-H (0.208 ± 0.010 mm) and CON groups. This result suggests that AA reduced the AD skin lesions formation on the mouse ears by reducing ear skin thickness.

### 3.3. Effects of AA on Immune System Organs in AD Mice

AD affects immune system organs by increasing the systemic immune response [19]. We measured the lymph node and spleen weights and sizes in the DNCB-treated AD animal model to evaluate the anti-AD effects of AA (Figure 4A,B). The lymph node and spleen sizes were larger in the AD-only group than in the CON group. In addition, the lymph nodes and spleen sizes in the AA-treated groups (AD + AA-L and AD + AA-H) were smaller than those in the AD-only group. The weight of the lymph nodes in the AD-only group (0.019 ± 0.001 g) was 4.8-fold higher compared with that in the CON group (0.004 ± 0.001 g). However, the weights of the lymph nodes in the AD + AA-L and AD + AA-H groups were (1.3-fold, respectively) reduced compared to those in the AD-only group (AD + AA-L: 0.015 ± 0.003 g, AD + AA-H: 0.015 ± 0.003 g; Figure 4B). In addition, the weights of the lymph nodes in the AA-H (0.014 ± 0.003 g) and AD + Pred (0.009 ± 0.004 g) groups were reduced by 1.1- and 1.4-fold, respectively, compared with those in the AD-only group. The spleen weight (0.106 ± 0.010 g) of the AD-only group was 1.1-fold higher than in the CON group (0.093 ± 0.010 g), which was not significant. However, the spleen weight in the AD + Pred group (0.073 ± 0.022 g) was 1.5-fold lower compared to that in the AD-only group, which was significant. In addition, the body weight was not significantly different between any group. These results suggest that oral AA treatment decreased lymph node and spleen weights in AD mice. Therefore, AA alleviates AD in mice.

### 3.4. AA Reduces Dermal and Epidermal Thickness, Collagen Deposition, and Mast Cell Count in an AD Animal Model

To examine the effects of AA on the DNCB-induced AD animal model, the dermal and epidermal thickness were evaluated using H&E staining (Figure 5). The epidermal thickness increased in the AD skin lesions through skin surface remodeling [20]. The thickness of the dorsal dermal and epidermal tissues of the AD-only group increased by 4.1- and 2.8-fold, respectively, than that of the CON group (Figure 5A–C). However, the dermal (1.4- and 1.7-fold) and epidermal (1.5- and 2.1-fold) thickness in the AD + AA-L and AD + AA-H groups were lower than that in the AD-only group. Interestingly, the dermal and epidermal thickness in the AD + AA-H group was reduced by 1.1- and 1.3-fold, respectively, than that of the AD + Pred group. Moreover, the dermal and epidermal thickness did not differ between the AA-H and CON groups. Thus, the dorsal dermal and epidermal thickness in DNCB-induced AD mice was regulated through AA administration.

The dermal and epidermal thickness in the ear tissue in the AD-only group was 2.1- and 3.4-fold thicker, respectively, than that in the CON group (Figure 5A,D,E). However, the dermal thickness in the ear was reduced in the AD + AA-L and AD + AA-H groups (2.0- and 2.6-fold, respectively) compared with that in the AD-only group. Moreover, the ear dermal thickness in the AD + AA-H group was reduced compared with that in the CON group (1.3-fold). Ear dermal thickness was also reduced in the AA-H group compared with that in the CON group; hence, we suggested that AA-H did not cause inflammation in normal tissue. Furthermore, the epidermal thicknesses in the AD + AA-L and AD + AA-H groups were 1.5- and 2.8-fold smaller, respectively, than those in the AD-only group. The dermal and epidermal thickness in the AD + AA-H group was increased compared with that in the AD + Pred group, but there was no difference between the two groups. Therefore, the dermal and epidermal thickness of the AD dorsal skin and ear decreased dose-dependently after AA treatment. This suggests that oral AA administration may inhibit AD skin lesions.

To determine the amount of collagen deposition and fibroid tissue in the dorsal skin of DNCB-induced AD mice, the background and collagen fibers were stained red and blue using Masson’s trichrome stain (Figure 5A,F,G). The number of collagen fibers in the AD-only group in the dorsal skin was more significantly increased (3.3-fold) than the CON group. AA treatment dose-dependently decreased the number of collagen fibers in the DNCB-induced AD animal model, which was confirmed by a decrease in the dermal thickness. AD + AA-L treatment decreased collagen fiber deposition (1.7-fold) compared with the AD-only group. Additionally, the number of collagen fiber in the AD + AA-H group was 1.9-fold lower compared with that in the AD-only group. Furthermore, collagen fiber deposition in the AD + AA-H group (1.9-fold) was significantly reduced in contrast with that in the AD + Pred group. There was no significant difference in collagen fiber deposition between the AA-H and CON groups. Hence, our results support that AA regulates AD skin lesions by inhibiting dermal hyperplasia and collagen overproduction.

To evaluate the effect of AA on mast cells, we stained the dorsal skin using toluidine blue in the DNCB-induced AD animal model (Figure 5A,D). Inflammatory mediators produced through mast cell activation caused allergic inflammation in AD; thus, mast cell control reduces inflammation in AD [21]. The mast cell count in the AD-only group was 18.7-fold greater than that in the CON group. In the AD + AA-L and AD + AA-H groups, mast cell infiltration decreased by 1.4- and 3.9-fold, respectively, compared with that in the AD-only group, while that in the AD + AA-H group was 2.2-fold lower than that in the AD + Pred group. In addition, there was no significant difference in mast cell infiltration between the AA-H and CON groups. Collectively, these results suggest that AA induces AD skin lesion recovery by reducing the dermal and epidermal thickness in the dorsal skin and ears and by reducing collagen deposition and mast cell infiltration in the dorsal skin.

### 3.5. Effects of AA on the Atopic Dermatitis-Related Genes in BALB/c AD Mice

We further evaluated the effects of AA on the levels of AD-related cytokines, including Th1- (TNF-α, IL-1β, and -12), Th2- (IL-4, -5, -6, -10, -13, and -31), and Th17-related (IL-17) cytokines (Figure 6A–J). TNF-α expression level in the AD-only group was 97% higher than in the CON group (Figure 6A). Nevertheless, the TNF-α level in the AD + AA-L and AD + AA-H groups was reduced (96% and 97%, respectively) compared with that of the AD-only group, and the value was not significantly different between the AD + AA-L, AD + AA-H, and CON groups. In addition, the TNF-α level decreased in the AD + AA-L (2%) and AD + AA-H (3%) groups in contrast with that in the AD + Pred group. The IL-1β expression in the AD group was increased by 74% compared with that in the CON group (Figure 6B). The IL-1β expression in the AD + AA-H group was decreased by 93% compared to that in the AD-only group. Interestingly, the IL-1β level in the AD + AA-H group was significantly decreased by 23% and 19% compared with that in the AD + Pred and CON groups, respectively. Furthermore, IL-12 expression in the AD-only group was upregulated by 81% compared with that in the CON group (Figure 6C). However, AA treatment (AD + AA-L and AD + AA-H groups) decreased IL-12 expression compared with that in the AD-only group (by 32% and 65%, respectively). Furthermore, the expression level of IL-12 in the AD + AA-H group was decreased by 34% compared with that in the AD + AA-L group. In addition, the IL-12 expression level was not significantly different among the AD + AA-H, AD + Pred, and CON groups (Figure 6C). Thus, AA treatment dose-dependently decreased Th1-related cytokine levels and inhibited DNCB-induced AD.

The mRNA expression levels of Th2-related cytokines are shown in Figure 6D–I. IL-4, -5, -6, -10, -31, and -13 expression levels were higher in the AD-only group (96%, 77%, 97%, 67%, 19%, and 100%, respectively) than in the CON group. IL-4, -5, -6, -13, and -31 were dose-dependently downregulated through AA treatment in AD mice. IL-4, -5, -6, -10, -31, and -13 expression in the AD + AA-L group was decreased by 71%, 62%, 82%, 14%, 49%, and 56%, respectively, compared with their expression in the AD-only group. Additionally, the IL-4, -5, -6, -10, -31, and -13 mRNA expression levels were suppressed in the AD + AA-H group (by 77%, 75%, 92%, 8%, 51%, and 89%, respectively) in contrast with the AD-only group. Moreover, IL-5, -6, and -13 expression in the AD + AA-H group was decreased by 2%, 23%, and 17%, respectively, compared with that in the AD + Pred group. IL-5 and -6 expression levels in the AD + AA-L (15% and 16%, respectively) and AD + AA-H (2% and 5%, respectively) groups were increased compared with those in the CON group; however, there was no significant difference. The IL-17 levels were not significantly different between any treatment group (Figure 6J). Therefore, AA treatment inhibits AD via inhibiting the Th1- and Th2-related cytokine mRNA expression in the AD animal model.

### 3.6. Effects of AA Treatment on Inflammation-Related Genes in the AD Animal Model

COX-2 and chemokine ligand 9 (CXCL9) mRNA expression was dose-dependently decreased after AA treatment (Figure 6K–L). AD skin lesions cause itching, which releases various pro-inflammatory cytokines from keratinocytes that exacerbate skin inflammation [22]. In the AD-only group, the COX-2 expression level was 56% higher than in the CON group; AA treatment (AD + AA-L and AD + AA-H groups) decreased COX-2 expression (47% and 59%, respectively) compared to the AD-only group. In addition, the expression of these molecules was not significantly different among the CON, AD + Pred, and AA groups (Figure 6K). CXCL9 expression in the AD-only group was increased by 16% compared with that in the CON group; however, there was no significant difference between the AD only and CON groups (Figure 6L). The CXCL9 expression level in the AD + AA-H treatment group was 38% lower than that in the AD-only group. Furthermore, CXCL9 expression was not significantly different between the AD + Pred and AD + AA-H groups. Thus, COX-2 and CXCL9 mRNA expression levels are regulated by AA treatment. This means that oral AA administration inhibits inflammatory cytokine production.

To demonstrate the effects of AA treatment on the mechanism of inflammation, we examined IL-6, TNF-α, COX-2, and IL-8 protein levels in BALB/c AD mice (Figure 7A). The IL-6 expression level in the AD-only group was increased by 92% compared with that in the CON group. However, IL-6 expression was decreased in the AD + AA-L and AD + AA-H groups by 45% and 66%, respectively, compared with that in the AD-only group. Moreover, TNF-α expression in the AD-only group was increased by 60% compared to that in the CON group. The TNF-α expression level after AA treatment (AD + AA-L and AD + AA-H groups) was decreased by 33% and 66%, respectively, compared with that in the AD-only group. Interestingly, TNF-α expression in the AD + AA-H group was reduced compared with that in the CON group (5%) (*p* < 0.05). COX-2 and IL-8 levels in the AD-only group increased by 36% and 58%, respectively, compared with those in the CON group, and those in the AD + AA-H group were (24% and 85%, respectively) lower than those in the AD-only group. IL-8 protein levels in the AD + AA-H and CON groups were not significantly different. IL-8 levels in the AD + AA-H group were downregulated by 7.8% compared with those in the AD + Pred group. Furthermore, AA treatment (AD + AA-L and AD + AA-H groups) concentration-dependently decreased the expression of IL-6, TNF-α, COX-2, and IL-8. These major findings suggest that AA treatment suppressed IL-6, TNF-α, COX-2, and IL-8 expression in vivo. Thus, we propose that AA administration inhibits AD lesions formation by downregulating the inflammatory pathways in DNCB-induced AD models.

### 3.7. Effects of AA Treatment on NF-κB, p-Akt, and MAPK Signaling in an AD Animal Model

The effects of AA treatment on the expression of NF-κB, *p*-Akt, and MAPK signaling in the dorsal tissues of AD mice were also determined (Figure 7B). The inhibition of NF-κB and MAPKs is critically related to the decrease of the anti-inflammatory response and mast cell count [23]. The NF-κB protein expression level in the AD-only group was increased (50%) compared with that of the CON group. The AA treatment dose-dependently reduced NF-κB protein expression (10.8% and 56.5%, respectively, in AD + AA-L and AD + AA-H groups) compared with that in the AD-only group. Furthermore, the NF-κB level in the AD + AA-H group was decreased by 6.6%, which was higher than that in the CON group. 

Interestingly, the NF-κB level in the AD + AA-H group (33%) was decreased compared with that in the AD + Pred group. The expression level of *p*-Akt protein in the AD-only group was increased by 44%, which was higher than that in the CON group. In the AA-treated groups (AD + AA-L and AD + AA-H groups), the *p*-Akt expression was decreased by 25% and 51%, respectively, compared with that in the AD-only group. Interestingly, the *p*-Akt expression level in the AD + AA-H group (7%) was downregulated compared with that in the CON group. The MAPK (*p*-p38, *p*-JNK, and *p*-ERK1/2) protein expression in the AD-only group was 40%, 24%, and 23% higher, respectively, compared with that in the CON group. Moreover, the AD + AA-L and AD + AA-H group of the *p*-p38 (25% and 52%, respectively), *p*-JNK (35% and 90%, respectively), and *p*-ERK1/2 (38% and 33%, respectively) levels were decreased compared with that in the AD-only group. NF-κB expression and MAPK signaling were dose-dependently decreased after AA treatment. Our results demonstrate that AA treatment inhibited allergic inflammation by restraining NF-κB, *p*-Akt, and MAPK signaling levels in the AD model.

## 4. Discussion

AD, which causes an increase in skin inflammation, is associated with various symptoms such as erythema, swelling, and cracked skin [3]. The expression of Th1- and Th2-related cytokines increases the severity of AD [3]. In this study, we evaluated whether AA treatment downregulates Th1- and Th2-related cytokine levels and reduces NF-κB, *p*-Akt, and MAPK signaling 1evels in a DNCB-induced AD animal model (Figure 6 and Figure 7). Our results showed that AA treatment reduces the number of mast cells, epidermal and dermal thickness, collagen deposition, and AD skin lesions formation in the ears and back of AD mice (Figure 3, Figure 4 and Figure 5). Thus, we initially focused on the therapeutic effects of AA treatment in inhibiting inflammation and modulating the expression of disease signaling pathways (Th1- and Th2-related cytokines) in DNCB-induced AD animal models.

Mast cells modulate the expression of keratinocyte adhesion proteins, pro-inflammatory cytokines, growth factors, and chemokines, which are produced by histamine [24]. Mast cells increase T-cell activation by modulating the function of naïve T-cells (Th1 and Th2) [25]. We confirmed that mast cell numbers decreased after AA treatment in inflammation-induced mice through DNCB treatment (Figure 5A,H). Furthermore, DNCB treatment enlarged the lymph nodes and spleen by causing inflammation [26]. In this study, AA treatment in the AD animal model decreased the lymph node and spleen weight compared to that in the CON group (Figure 4). Mast cells also activate matrix metalloproteinases through induction of collagen deposition; thus, it is known as a pro-fibrotic [27]. Our results showed that collagen deposition was upregulated in the AD-only group. However, AA treatment dose-dependently reduced collagen fiber deposition and epidermal and dermal thickness (Figure 5). Our results are consistent with those of a previous study showing that *Solanum nigrum* L. treatment downregulates mast cell infiltration and epidermal and dermal thickness in a DNCB-induced AD animal model [28]. Thus, AA treatment regulated mast cells infiltration and skin lesion formation in AD, which might explain the recovery from AD skin lesions.

CD4^+^ T-cells, which play important roles in the immune response, have been classified into Th1, Th2, Treg, and Th17 cells [6]. Each Th cell type produces different cytokines by controlling cell function and immunity [29]. In particular, Th1 cells cause immune diseases, and Th2 cells mediate allergy and asthma [30]. AD disrupts the immunological balance between Th1 and Th2 cells in inflammation-induced skin lesions [31]. However, the Th1 and Th2 cell balance can be confirmed by the IgG2/IgG1 ratio, as a Th1 and Th2 lymphocyte marker [31]. This study evaluated whether AA modulates the immunological balance between Th1 and Th2 cells and inhibits the development of DNCB-induced AD (Figure 6). In addition to inducing isotype conversion to IgE, Th2 cells play a significant role in increasing inflammation by enhancing the expression of adhesion molecules in endothelial cells [30]. Therefore, the inhibition of Th2 activation can reduce the skin symptoms of AD [32]. 

Furthermore, mast cell differentiation can progress by overproducing AD skin lesions through IL-4 and IL-13 expression [3,5]. Th1 cells produce effector cytokines, including TNF-α, IFN-γ, IL-12, IL-2, and TGF-β1; thus, downregulating Th1-related cytokines can inhibit the development of chronic AD [3]. Our results showed that AA treatment effectively regulates the enhancement of Th1- and Th2-related pathways, thus, downregulating the immune response in the AD model (Figure 6). In addition, our previous results have indicated that *C. asiatica* treatment inhibits the expression of Th1- (TNF-α) and Th2-related (IL-4, -5, -6, -13, and -31) cytokines in a DNCB-induced AD animal model, thereby reducing the formation of AD skin lesions [15]. However, our results confirmed the hypothesis that AA treatment diminishes IL-13 mRNA expression and also reduces Th1- and Th2-associated cytokine levels in DNCB-induced AD mice. IL-13, the main mediator of allergic inflammation, is expressed in both acute and chronic AD [6]. 

Peripheral neurons stimulate IL-4 and -13 expression, which causes itching [33]. IL-13 reduces collagen degradation in human dermal fibroblasts, which causes moisture loss in the skin through reduced MMP-13 expression and causes fibrosis due to excessive collagen deposition along with dermal thickening of AD skin lesions [33]. In this study, AA treatment suppressed collagen deposition and IL-13 mRNA expression levels (Figure 5F,G and Figure 6H). Furthermore, chronic AD has been characterized to upregulate Th1-related pathways and influence tissue remodeling by increasing collagen deposition and dermal thickness [34]. Similarly, we determined that the dermal and epidermal thickness in mice in the AD + AA-H group was notably diminished compared to that in the AD-only group (Figure 5B,C). Therefore, we suggest that AA has an immunomodulatory effect through maintenance of the moisture content in the skin and reduction of itchiness, thereby inhibiting AD skin lesion formation.

Pro-inflammatory cytokines, including IL-1β, -4, -6, -13, and TNF-α, regulate inflammatory skin lesions and the immune response in AD [35]. Our results suggest that AA downregulated IL-6, -1β, and TNF-α expression in AD mice (Figure 6). COX-2 is involved in immune functions and the inflammatory response, which play an important role in the conversion of arachidonic to prostaglandins in the skin [35]. Our previous study demonstrated that *C. asiatica* treatment downregulates inflammation-related genes, such as TNF-α, COX-2, and iNOS, in a DNCB-treated animal model [15]. Similarly, AA treatment inhibited the expression levels of COX-2, CXCL9, IL-6, TNF-α, and IL-8 in this study (Figure 7A). Regulation of COX-2 expression led to a downregulated inflammatory response in a DNCB-induced animal model [34]. Hence, we conclude that AA can inhibit the expression of anti-inflammatory cytokines.

Skin stimulation agents, including LPS, TNF-α, or DNCB, act as irritants to keratinocytes, and these stimulated keratinocytes, in turn, activate the MAPK, Akt, and NF-κB signaling pathways [36]. The *p*-p38 pathway is more responsive to oxidative stress than other MAPK signaling pathways (such as p-ERK and p-JNK) [37], and the *p*-p38 pathway is involved in the chronic phase of AD pathogenesis [38]. NF-κB is a transcription factor that facilitates allergic diseases, such as AD and asthma, by increasing the levels of inflammatory cytokines [39,40]. Furthermore, the PI3K/Akt pathway regulates the immune reaction of eosinophils, T and B lymphocytes, and mast cells [41]. In LPS-induced macrophages, AA inhibits IL-6, -1β, and TNF-α expression via the p38, ERK1/2, JNK, and NF-κB pathways [42]. This study showed that AA reduced the expression of NF-κB, *p*-Akt, and MAPK signaling pathways in DNCB-induced AD mice (Figure 7). Thus, our results suggest that AA treatment reduces the inflammation-related gene by downregulating the NF-κB and MAPK signaling pathways.

## 5. Conclusions

This is the first study to investigate the immunomodulatory and anti-inflammatory effects of AA in an AD animal model. We suggest that 20 µg/mL AA downregulates inflammatory cytokines in TNF-α + IFN-γ-treated HaCaT cells. The AA treatment for 18 days regulated swelling, excoriation, edema, and scarring in the ears and dorsal skin in DNCB-induced AD mice. AA administration also reduced lymph and spleen weight, dermal and epidermal thickness, collagen deposition, and mast cell count in a DNCB-induced AD animal model, thereby reducing the formation of skin lesions due to allergic reactions. Furthermore, AA was proved to have immunomodulatory and anti-inflammatory effects as it blocked the activity of Th1- and Th2-related cytokines. In addition, the levels of inflammatory cytokines as well as the activity of NF-κB, *p*-Akt, and MAPK signaling pathways were suppressed in the AD + AA-H group. In conclusion, our results suggest that AA administration reduces the formation of AD skin lesions by enhancing immunomodulation and inhibiting inflammation and might be a potent therapy for attenuating AD symptoms such as itching, swelling, crust formation, and leathery skin.

## Figures and Tables

**Figure 1 nutrients-13-02448-f001:**
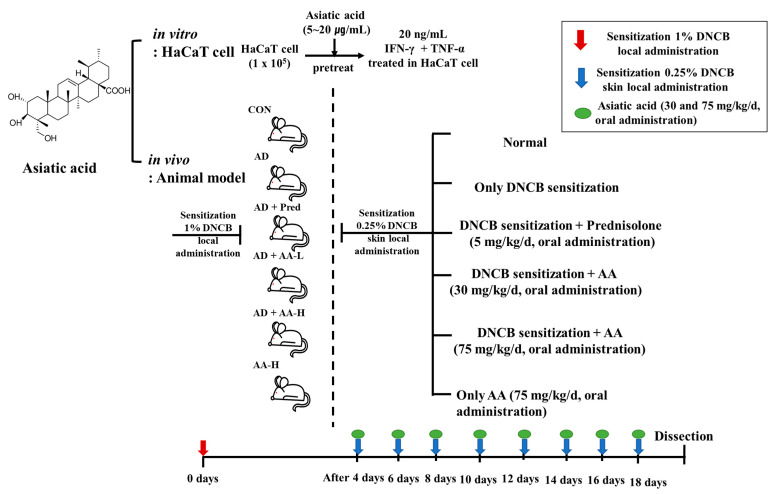
To determine the effect of asiatic acid (AA) in vitro in 20 ng/mL IFN-γ + TNF-α-treated HaCaT cells pretreated with 5–20 μg/mL AA and in a 1% 2,4-dinitrochlorobenzene (DNCB)-induced atopic dermatitis (AD) animal model. AD mice were randomly divided into six groups (*n* = 7 per group). DNCB promoted AD skin lesions in the mice. Briefly, 200 µL 0.25% DNCB was applied to each mouse’s ear and, after 4 d, application was repeated once every 2 d for 18 d. AA (30 and 75 mg/kg/d) was orally administered. The AA-H mice were orally administered 75 mg/kg/d AA. Prednisolone was orally administered (5 mg/kg/d).

**Figure 2 nutrients-13-02448-f002:**
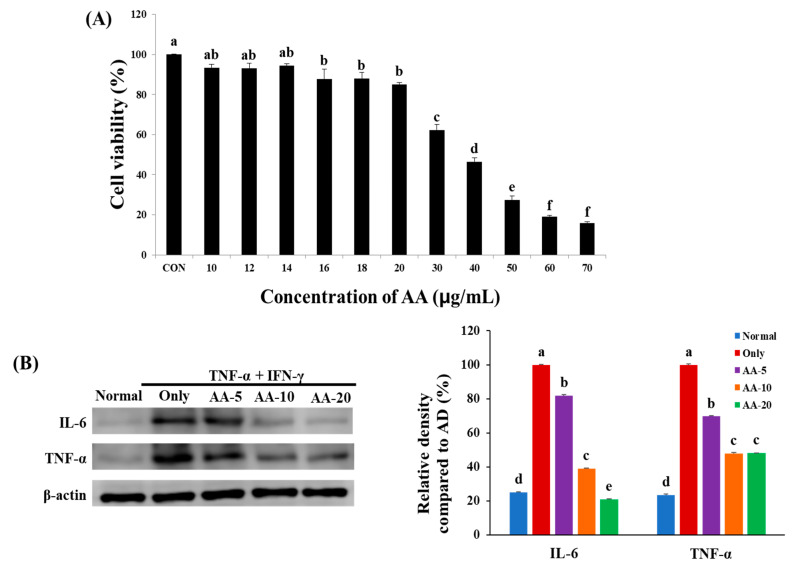
To evaluate the effect of asiatic acid (AA), (**A**) cell viability in HaCaT cells was examined using the 3-(4,5-dimethylthiazol-2-yl)-2,5-diphenyltetazolium bromide assay. CON, control. (**B**) Effects of AA on the inflammation mechanism of tumor necrosis factor (TNF)-α- and interferon (IFN)-γ-treated HaCaT cells. According to Duncan’s multiple range test, significant values are indicated by different letters (*p* < 0.05), and values express the mean ± standard deviation (*n* = 3). Normal, control cells; Only, 20 ng/mL TNF-α- and IFN-γ-treated cells; AA-5, -10, and -20, cells pretreated with 5, 10, and 20 µg/mL AA, respectively, and then treated with 20 ng/mL TNF-α and IFN-γ.

**Figure 3 nutrients-13-02448-f003:**
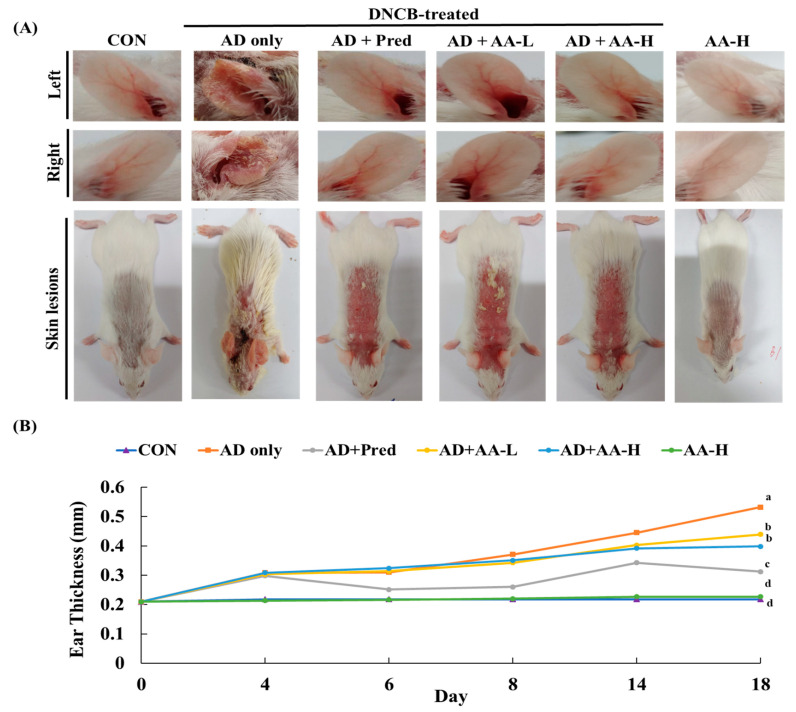
To determine the effect of asiatic acid (AA), (**A**) the ears and dorsal skin were photographed on day 18 after treatment. (**B**) Ear thickness was analyzed by a dial thickness gauge 24 h after 2,4-dinitrochlorobenzene (DNCB) treatment. The ear thickness was measured for 18 days, and different letters indicate significantly different values according to Duncan’s multiple range test (*p* < 0.05) (*n* = 7 per group). The values are expressed as mean ± standard deviation. CON, control group; AD only, mice sensitized with 0.25% DNCB in 200 µL 3:1 acetone/bean oil; AD + AA-L and AD + AA-H, 30 and 75 mg/kg AA-treated DNCB-sensitized mice, respectively, in 100 µL of sterilized drinking water; AA-H, mice orally administered 75 mg/kg AA; AD + Pred, positive control mice treated with 5 mg/kg prednisolone in 100 µL of sterilized drinking water.

**Figure 4 nutrients-13-02448-f004:**
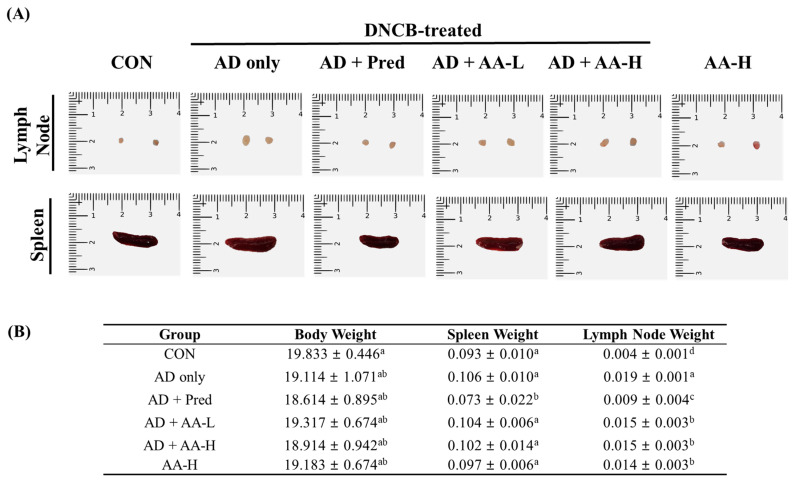
Effect of asiatic acid (AA) on immune organs in the atopic dermatitis (AD) animal model. (**A**) The lymph nodes and spleens of the AD mice were photographed on day 18 after treatment. (**B**) Descriptive statistics showing differences in body, lymph node, and spleen weights in each group. The significant values according to Duncan’s multiple range test (*p* < 0.05) are indicated by different letters, and values are expressed as the mean ± standard deviation (*n* = 7 per group). CON, control group; AD only, mice sensitized with 0.25% 2,4-dinitrochlorobenzene (DNCB) in 200 µL of 3:1 acetone/bean oil; AD + AA-L and AD + AA-H, 30 and 75 mg/kg AA-treated DNCB-sensitized mice, respectively, in 100 µL of sterilized drinking water; AA-H, mice orally administered 75 mg/kg AA; AD + Pred, positive control mice treated with 5 mg/kg prednisolone in 100 µL of sterilized drinking water.

**Figure 5 nutrients-13-02448-f005:**
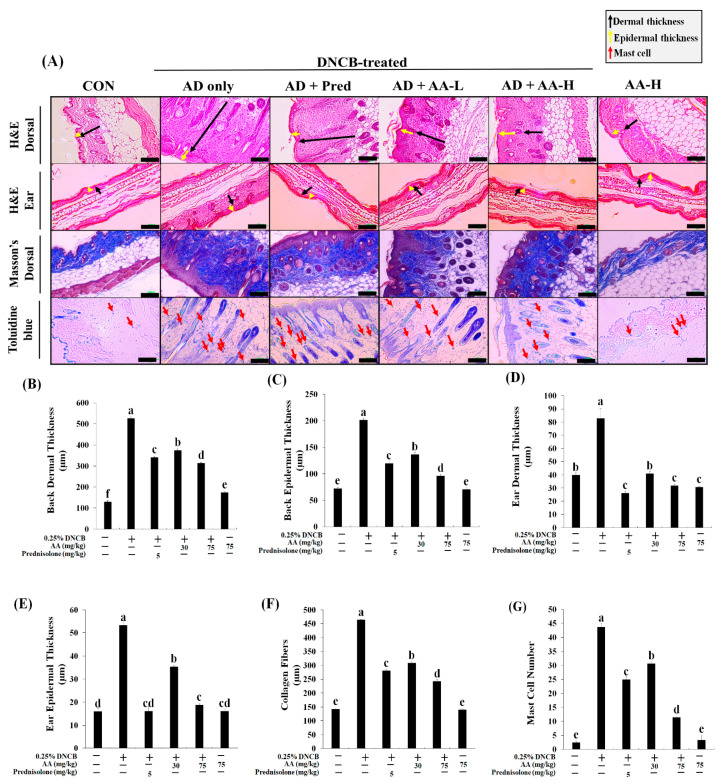
Effects of asiatic acid (AA) on dermal and epidermal thickness, collagen fibers, and mast cell infiltration were assessed histologically. (**A**) Atopic dermatitis (AD) skin lesions were stained using hematoxylin and eosin (H&E), Masson’s trichrome, or toluidine blue, and were photographed under a light microscope at 100× magnification (scale bar: 100 µm). (**B**–**E**) The dermal and epidermal thicknesses were evaluated using H&E-stained dorsal and ear tissue microphotographs. (**F**) Collagen fibers were evaluated using Masson’s trichrome staining. (**G**) Mast cells were counted after toluidine blue staining; the blue points denote mast cells. Values were significant according to Duncan’s multiple range test (*p* < 0.05), and the values are represented as the mean ± standard deviation (*n* = 7 per group). The significant values are represented by different letters *(p <* 0.05). CON, control group; AD only, mice sensitized with 0.25% 2,4-dinitrochlorobenzene (DNCB) in 200 µL of 3:1 acetone/bean oil; AD + AA-L and AD + AA-H, 30 and 75 mg/kg AA-treated DNCB-sensitized mice, respectively, in 100 µL of sterilized drinking water; AA-H, mice orally administered 75 mg/kg AA; AD + Pred, positive control mice treated with 5 mg/kg prednisolone in 100 µL of sterilized drinking water.

**Figure 6 nutrients-13-02448-f006:**
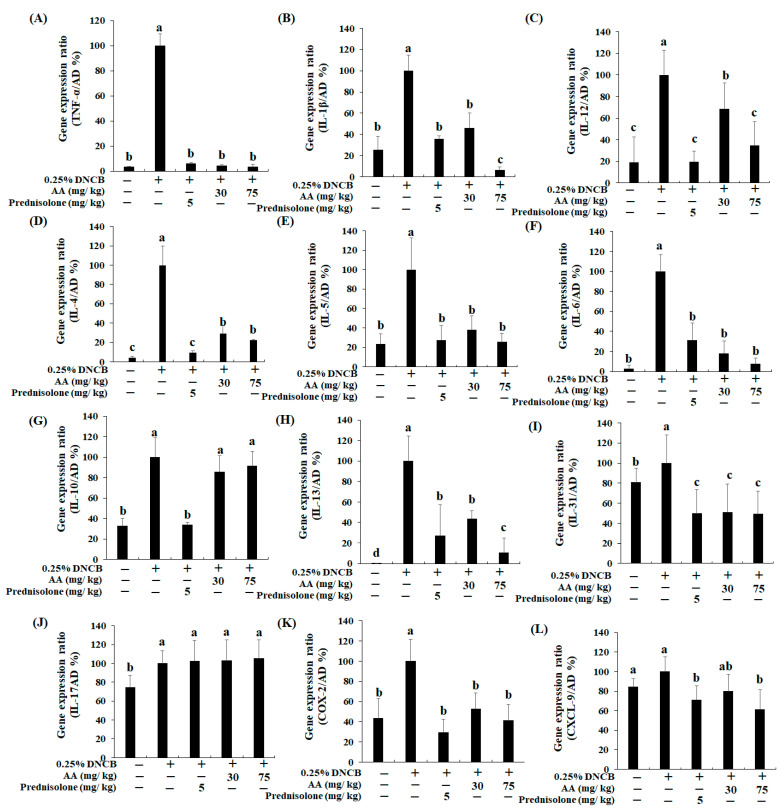
Asiatic acid (AA) effects on atopic dermatitis (AD)-related cytokine expression level in the dorsal skin of 2,4-dinitrochlorobenzene (DNCB)-induced AD mice. (**A**) Tumor necrosis factor (TNF)-α; (**B**) interleukin (IL)-1β; (**C**) IL-12; (**D**) IL-4; (**E**) IL-5; (**F**) IL-6; (**G**) IL-10; (**H**) IL-13; (**I**) IL-31; (**J**) IL-17; (**K**) cyclooxygenase (COX)-2; and (**L**) chemokine ligand (CXCL)-9. As per Duncan’s multiple range test (*p* < 0.05), the significant values are represented by different letters. The results are expressed as the mean ± standard deviation (*n* = 7 per group).

**Figure 7 nutrients-13-02448-f007:**
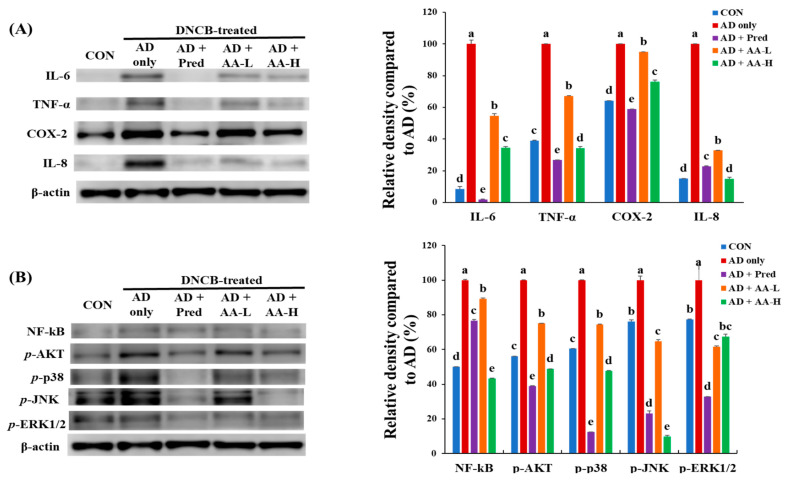
Effects of asiatic acid (AA) on the expression of inflammatory and signaling proteins in the dorsal skin. Expression of (**A**) anti-inflammatory pathway proteins and (**B**) nuclear factor kappa B (NF-κB), phosphor(*p*)-protein kinase B (Akt), *p*-p38, *p*-c-Jun N-terminal kinase (JNK), and *p*-Extracellular-Regulated Protein Kinase 1/2 (ERK1/2) proteins. Different letters indicate significant values per Duncan’s multiple range test (*p* < 0.05), and values are expressed as mean ± standard deviation (n = 7 per group). CON, control group; AD only, mice sensitized with 0.25% 2,4-dinitrochlorobenzene (DNCB) in 200 µL of 3:1 acetone/bean oil; AD + AA-L and AD + AA-H, 30 and 75 mg/kg AA-treated DNCB-sensitized mice, respectively, in 100 µL of sterilized drinking water; AD + Pred, positive control mice treated with 5 mg/kg prednisolone in 100 µL of sterilized drinking water.

## Data Availability

The datasets generated during and/or analyzed during the current study are available from the corresponding author on reasonable request.

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
