# Peer review of "Immunomodulatory and Anti-inflammatory Effects of Asiatic Acid in a DNCB-Induced Atopic Dermatitis Animal Model"

_nutrients, 2021, doi:10.3390/nu13072448_

Round 1
Reviewer 1 Report
Asiatic acid is a well-known natural product isolated from various natural resources, due to its potential as a bioactive metabolite, there are numerous pharmacological applications that have been studied in the last 10 years. The submitted manuscript contributes to further research on the biomedical potential of this compound. The authors have done a good research job of AD in vivo obtaining results consistent with previous antecedents. Therefore, I consider it acceptable for publication with minor revisión.
1.- I suggest to the authors a meticulous review of the references, especially in the introduction, where I have detected that the information indicated does not correspond to the cited reference. (For example, line 48, it is not reference 8, it would be reference 9, this happens more often).
2.- It is necessary to improve the image quality of the formula and in the manuscript. In fig 1 the formula is deformed and the text boxes have poor resolution. In fig 3 The image is deformed, wider than it is tall. Fig 5 is also deformed, but the worst thing is that the letter in the diagrams is so small that it is impossible to read it and the axis numbers are not visible.
Author Response
Response Letter to Reviewer’s Comments
Manuscript ID: nutrients-1271351
Manuscript title: Immunomodulatory and Anti-inflammatory Effects of Asiatic Acid in a DNCB-induced Atopic Dermatitis Animal Model
Response to comments of Reviewer #1
Comments and Suggestions for Authors
Asiatic acid is a well-known natural product isolated from various natural resources, due to its potential as a bioactive metabolite, there are numerous pharmacological applications that have been studied in the last 10 years. The submitted manuscript contributes to further research on the biomedical potential of this compound. The authors have done a good research job of AD in vivo obtaining results consistent with previous antecedents. Therefore, I consider it acceptable for publication with minor revisión.
- I suggest to the authors a meticulous review of the references, especially in the introduction, where I have detected that the information indicated does not correspond to the cited reference. (For example, line 48, it is not reference 8, it would be reference 9, this happens more often).
- Thank you for your valuable comments. We have revised the reference section accordingly.
Page 14:
- Meeran, M.F.N.; Goyal, S.N.; Suchal, K.; Sharma, C.; Patil, C.R.; Ojha, S.K. Pharmacological properties, molecular mechanisms, and phar-maceutical development of asiatic acid: A pentacyclic triterpenoid of therapeutic promise. Front. Pharmacol. 2018, 9, 892. doi:10.3389/fphar.2018.00892.
- Acebey-Castellon, IL.; Voutquenne-Nazabadioko, L.; Doan Thi Mai, H.; Roseau, N.; Bouthagane, N.; Muhammad, D.; Le Magrex Deba,r, E.; Gamgloff, SC.; Litaudon, M.; Sevenet, T.; et al. Triterpenoid saponins from Symplocos lancifolia. Journal of natural products. 2011. 74, 163-168. doi: 10.1021/np100502y.
- Pakdeechote, P.; Bunbupha, S.; Kukongviriyapan, U.; Prachaney, P.; Khrisanapant, W.; Kukongviriyapan, V. Asiatic acid alleviates hemody-namic and metabolic alterations via restoring eNOS/iNOS expression, oxidative stress, and inflammation in diet-induced metabolic syndrome rats. Nutrients. 2014, 6, 355–370. doi:10.3390/nu6010355.
- Gou, X.J.; Bai, H.H.; Liu, L.W.; Chen, H.Y.; Shi, Q.; Chang, L.S.; Ding, M.M.; Shi, Q.; Zhou, M.X.; Chen, W.L.; et al. Asiatic Acid interferes with invasion and proliferation of breast cancer cells by inhibiting WAVE3 activation through PI3K/AKT signaling pathway. Biomed Res. Int. 2020, 2020, 1874387. doi:10.1155/2020/1874387.
- Huang, S.S.; Chiu, C.S.; Chen, H.J.; Hou, W.C.; Sheu, M.J.; Lin, Y.C.; Shie, P.H. Antinociceptive activities and the mechanisms of an-ti-inflammation of asiatic acid in mice. Evid-based Complement Alternat Med. 2011, 2011, 895857. doi:10.1155/2011/895857.
- Li, Y.; Yang, F.; Yuan, M.; Jiang, L.; Yuan, L.; Zhang, X.; Li, Y.; Dong, L.; Bao, X.; Yin, S. Synthesis and evaluation of asiatic acid derivatives as anti-fibrotic agents: Structure/activity studies. Steroids. 2015, 96, 44–49. doi:10.1016/j.steroids.2014.11.001.
- Hao, C.; Wu, B.; Hou, Z.; Xie, Q.; Liao, T.; Wang, T.; Ma, D. Asiatic acid inhibits LPS-induced inflammatory response in human gingival fibroblasts. Int. Immunopharmacol. 2017, 50, 313–318. doi:10.1016/j.intimp.2017.07.005.
- Lee, Y.; Choi, H.K.; N'deh K.P.U.; Choi, Y.J.; Fan, M.; Kim, E.K..; Chung, K.H.; An, J.H. Inhibitory effect of Centella asiatica extract on DNCB-induced atopic dermatitis in HaCaT cells and BALB/c mice. Nutrients. 2020, 12, 411. doi: 10.3390/nu12020411.
- Hwang, Y.S.; Chang, B.Y.; Kim, T.Y.; Kim, S.Y. Ameliorative effects of green tea extract from tannase digests on house dust mite anti-gen-induced atopic dermatitis-like lesions in NC/Nga mice. Arch. Dermatol. Res. 2019, 311, 109–120. doi:10.1007/s00403-018-01886-6.
- Choi, E.J.; Iwasa, M.; Han, K. I; Kim, W.J.; Tang, Y.; Hwang, Y.J.; Chae, J.R.; Han, W.C.; Shin, Y.S.; Kim, E.K. Heat-killed enterococcus faecalis EF-2001 ameliorates atopic dermatitis in a murine model. Nutrients. 2016, 8, 146. doi:10.3390/nu8030146.
- Boniface, K.; Bernard, F.X.; Garcia, M.; Gurney, A.L.; Lecron, J.C.; Morel, F. IL-22 inhibits epidermal differentiation and induces proin-flammatory gene expression and migration of human keratinocytes. J. Immunol. 2005, 174, 3695–3702. doi:10.4049/jimmunol.174.6.3695.
- Yang, H.R.; Lee, H.; Kim, J.H.; Hong, I.H.; Hwang, D.H.; Rho, I.R.; Kim, G.S.; Kim, E.; Kang, C. Therapeutic effect of Rumex japonicus Houtt. on DNCB-induced atopic dermatitis-like skin lesions in Balb/c mice and human keratinocyte HaCaT cells. Nutrients. 2019, 11, 573. doi:10.3390/nu11030573.
- De Vuyst, E.; Salmon, M.; Evrard, C.; Lambert de Rouvroit, C.; Poumay, Y. Atopic Dermatitis studies through in vitro models. Front. Med. 2017, 4, 119. doi:10.3389/fmed.2017.00119.
- Galli, S.J.; Tsai, M. IgE and mast cells in allergic disease. Nat. Med. 2012, 18, 693–704. doi:10.1038/nm.2755.
- Lee, J.H.; Jeon, Y.D.; Lee, Y.M.; Kim, D.K. The suppressive effect of puerarin on atopic dermatitis-like skin lesions through regulation of inflammatory mediators in vitro and in vivo. Biochem Biophys Res Commun. 2018, 498, 707–714. doi:10.1016/j.bbrc.2018.03.018.
- Mechesso, A.F.; Lee, S.J.; Park, N.H.; Kim, J.Y.; Im, Z.E.; Suh, J.W.; Park, S.C. Preventive effects of a novel herbal mixture on atopic der-matitis-like skin lesions in BALB/C mice. BMC Complement. Altern. Med. 2019, 19, 1–13. doi:10.1186/s12906-018-2426-z.
- Liu, F.T.; Goodarzi, H.; Chen, H.Y. IgE, mast cells, and eosinophils in atopic dermatitis. Clin Rev Allergy Immunol. 2011, 41, 298–310. doi:10.1007/s12016-011-8252-4.
- Nakae, S.; Suto, H.; Iikura, M.; Kakurai, M.; Sedgwick, J.D.; Tsai, M.; Galli, S.J. Mast cells enhance T cell activation: importance of mast cell costimulatory molecules and secreted TNF. J. Immunol. 2006, 176, 2238–2248. doi:10.4049/jimmunol.176.4.2238.
- Shea-Donohue, T.; Stiltz, J.; Zhao, A.; Notari, L. Mast cells. Curr Gastroenterol Rep. 2010, 12, 349–357. doi:10.1007/s11894-010-0132-1.
- Piera, L.; Olczak, S.; Kun, T.; Galdyszynska, M.; Ciosek, J.; Szymanski, J.; Drobnik, J. Disruption of histamine/H3 receptor signal reduces collagen deposition in cultures scar myofibroblasts. J Physiol Pharmacol. 2019, 70, 239–247. doi:10.26402/jpp.2019.2.07.
- Hong, S.; Lee, B.; Kim, J.H.; Kim, E.Y.; Kim, M.; Kwon, B.; Cho, H.R.; Sohn, Y.; Jung, H.S. Solanum nigrum Linne improves DNCB-induced atopic dermatitis-like skin disease in BALB/c mice. Mol. Med. Rep. 2020, 22, 2878–2886. doi:10.3892/mmr.2020.11381.
- Puniya, B.L.; Todd, R.G.; Mohammed, A.; Brown, D.M.; Barberis, M.; Helikar, T. A mechanistic computational model reveals that plasticity of CD4+ T cell differentiation is a function of cytokine composition and dosage. Front. Physiol. 2018, 9, 878. doi:10.3389/fphys.2018.00878.
- Hamid, Q.; Boguniewicz, M.; Leung, D.Y. Differential in situ cytokine gene expression in acute versus chronic atopic dermatitis. J.Clin. Invest. 1994, 94, 870–876. doi: 10.1172/JCI117408.
- Szegedi, A.; Baráth, S.; Nagy, G.; Szodoray, P.; Gál, M.; Sipka, S.; Bagdi, E.; Banham, A.H.; Krenács, L. Regulatory T cells in atopic dermatitis: epidermal dendritic cell clusters may contribute to their local expansion. Br. J. Dermatol. 2009, 160, 984–993. doi:10.1111/j.1365-2133.2009.09035.x.
- Huang, W.C.; Huang, C.H.; Hu, S.; Peng, H.L.; Wu, S.J. Topical spilanthol inhibits MAPK signaling and ameliorates allergic inflammation in DNCB-induced atopic dermatitis in mice. Int. J. Mol. Sci. 2019, 20, 2490. doi:10.3390/ijms20102490.
- Bieber, T. Interleukin-13: Targeting an underestimated cytokine in atopic dermatitis. Allergy. 2020, 75, 54–62. doi:10.1111/all.13954.
- Gil, T.Y.; Kang, Y.M.; Eom, Y.J.; Hong, C.H.; An, H.J. Anti-atopic dermatitis effect of Seaweed Fulvescens extract via inhibiting the STAT1 pathway. Mediators Inflamm. 2019, 2019, 3760934. doi:10.1155/2019/3760934.
- Lee, Y.J.; Oh, M.J.; Lee, D.H.; Lee, Y.S.; Lee, J.; Kim, D.H.; Choi, C.H.; Song, M.J.; Song, H.S.; Hong, J.T. Anti-inflammatory effect of bee venom in phthalic anhydride-induced atopic dermatitis animal model. Inflammopharmacology. 2020, 28, 253–263. doi:10.1007/s10787-019-00646-w.
- Herlaar, E.; Brown, Z. p38 MAPK signalling cascades in inflammatory disease. Mol. Med. Today. 1999, 5, 439–447. doi:10.1016/S1357-4310(99)01544-0.
- Yi, C.; Si, L.; Xu, J.; Yang, J.; Wang, Q.; Wang, X. Effect and mechanism of asiatic acid on autophagy in myocardial ischemia‑reperfusion injury. Exp. Ther. Med. 2020, 20, 54. doi:10.3892/etm.2020.9182.
- Johansen, C.; Kragballe, K.; Westergaard, M.; Henningsen, J.; Kristiansen, K.; Iversen, L. The mitogen-activated protein kinases p38 and ERK1/2 are increased in lesional psoriatic skin. Br. J. Dermatol. 2005, 152, 37–42. doi:10.1111/j.1365-2133.2004.06304.x.
- Qin, W.; Duan, J.; Xie, X.; Kang, J.; Deng, T.; Chen, M. Exposure to diisononyl phthalate promotes atopic march by activating of NF-κB and p38 MAPK. Toxicol. Appl. Pharmacol. 2020, 395, 114981. doi:10.1016/j.taap.2020.114981.
- Ye, J.; Piao, H.; Jiang, J.; Jin, G.; Zheng, M.; Yang, J.; Jin, X.; Sun, T.; Choi, Y.H.; Li, L.; et al. Polydatin inhibits mast cell-mediated allergic inflammation by targeting PI3K/Akt, MAPK, NF-κB and Nrf2/HO-1 pathways. Sci. Rep. 2017, 7, 1–13. doi:10.1038/s41598-017-12252-3.
- Chen, L.; Chen, J.; Xie, C.M.; Zhao, Y.; Wang, X.; Zhang, Y.H. Maternal Disononyl Phthalate Exposure Activates Allergic Airway Inflam-mation via Stimulating the Phosphoinositide 3-kinase/Akt Pathway in Rat Pups. Biomed. Environ. Sci. 2015, 28, 190–198. doi:10.3967/bes2015.025.
- Yun, K.J.; Kim, J.Y.; Kim, J. B; Lee, K.W.; Jeong, S.Y.; Park, H.J.; Jung, H.J.; Cho, Y.W.; Yun, K.; Lee, K.T. Inhibition of LPS-induced NO and PGE2 production by asiatic acid via NF-κB inactivation in RAW 264.7 macrophages: possible involvement of the IKK and MAPK pathways. Int. Immunopharmacol. 2008, 8, 431–441. doi:10.1016/j.intimp.2007.11.003.
- It is necessary to improve the image quality of the formula and in the manuscript. In fig 1 the formula is deformed and the text boxes have poor resolution. In fig 3 The image is deformed, wider than it is tall. Fig 5 is also deformed, but the worst thing is that the letter in the diagrams is so small that it is impossible to read it and the axis numbers are not visible.
- Thank you for the correction, we have changed the poor-resolution (Fig.1,3, and 5) picture with clearer images below.
Page 3:
Page 6:
Page 8:

Reviewer 2 Report
This manuscript describes the immunomodulatory and anti-inflammatory effects of Asiatic acid in DNCB-induced AD model. However, it still can be improved.
In the abstract and manuscript, please add the 2, 4-Dinitrocholrlbenzene (DNCB) when using the abbreviation DNCB in the first time.
It might will be good if you add a figure for the structure of AA in the Introduction part.
Please also mention n=? information for each of figure as the author only mention in figure 1.
Why don’t you add “*” to label the statically significant values (p < 0.05)
Please add detail information for each figure. Such as in figure 2C, in the top of each bar was labeled by different letter?
A very small correction, by adding the space between symbols, will need to fix throughout the whole manuscript, e.g. n = 7 per group
Some of the English is rather awkward in places. Hereby, I suggest the author should read the manuscript again carefully.
How do the author choose the lower and higher dose (30 and 75 mg/kg/d) and the positive control using for 5 mg/kg prednisolone?
I also encourage the authors to address the comments below.
Please change the reference 8 to some chemistry references.
P2L55 the number 4 of the word CCl4 should be lowercase.
In figure 5A, would it be possible to use different arrow to point out the epidermal and dermal thickness to show the difference?
Why don’t the author test the protein expression of ERK1/2 and JNK?
Author Response
Response to comments of Reviewer #2
Comments and Suggestions for Authors
This manuscript describes the immunomodulatory and anti-inflammatory effects of Asiatic acid in DNCB-induced AD model. However, it still can be improved.
- In the abstract and manuscript, please add the 2, 4-Dinitrocholrlbenzene (DNCB) when using the abbreviation DNCB in the first time.
- Thank you for your valuable comments. We have revised the Abstract section accordingly. Also, we have double-checked English grammar.
Page 1:
The 2,4-dinitrocholrlbenzene (DNCB)-induced AD animal model was developed by administering two AA concentrations (30 and 75 mg/kg/d: AD + AA-L and AD + AA-H groups, respectively) for 18 days.
- It might be good if you add a figure for the structure of AA in the Introduction part.
- Since the Asiatic acid chemical structure was shown in Figure 1, we just need to add some Asiatic acid chemical information in introduction.
Page 2:
AA is a pentacyclic triterpenoid that is substituted by a carboxy group at posi-tion 28 and hydroxy groups at positions 2, 3, and 23 (the 2α, 3β stereoisomer) [9].
- Please also mention n=? information for each of figure as the author only mention in figure 1.
- We added n=? to the figure.
Page 3:
Figure 1. To determine the effect of asiatic acid (AA) (A) in vitro in 20 ng/mL IFN-γ + TNF-α-treated HaCaT cells pre-treated with 5–20 μg/mL AA and (B) in a 1% 2,4-dinitrochlorobenzene (DNCB)-induced atopic dermatitis (AD) animal model. AD mice were randomly divided into six groups (n = 7 per group). DNCB promoted AD skin lesions in the mice. Briefly, 200 µL 0.25% DNCB was applied to each ear of each mouse and, after 4 d, application was repeated once every 2 days for 18 days. AA (30 and 75 mg/kg/d) was orally administered. The AA-H mice were orally administered 75 mg/kg/d AA. Prednisolone was orally administered (5 mg/kg/d).
Page 5:
Figure 2. To evaluate the effect of asiatic acid (AA), (A) cell viability in HaCaT cells was examined using the 3-(4,5-dimethylthiazol-2-yl)-2,5-diphenyltetazolium bromide assay. CON, control. (B) Effects of AA on the inflammation mechanism of tumor necrosis factor (TNF)-α- and interferon (IFN)-γ-treated HaCaT cells. According to Duncan’s multiple range test, significant values are indicated by different letters (p < 0.05), and values express the mean ± standard deviation (n = 3). Normal, control cells; Only, 20 ng/mL TNF-α- and IFN-γ-treated cells; AA-5, -10, and -20, cells pretreated with 5, 10, and 20 µg/mL AA, respectively, and then treated with 20 ng/mL TNF-α and IFN-γ.
Page 6:
Figure 3. To determine the effect of asiatic acid (AA), (A) the ears and dorsal skin were photographed on day 18 after treatment. (B) Ear thickness was analyzed by a dial thickness gauge 24 h after 2,4-dinitrochlorobenzene (DNCB) treatment. The ear thickness was measured for 18 days, and different letters indicate significantly different values according to Duncan's multiple range test (p < 0.05) (n = 7 per group). The values are expressed as mean ± standard deviation. CON, control group; AD only, mice sensitized with 0.25% DNCB in 200 µL 3:1 acetone/bean oil; AD + AA-L and AD + AA-H, 30 and 75 mg/kg AA-treated DNCB-sensitized mice, respectively, in 100 µL of sterilized drinking water; AA-H, mice orally administered 75 mg/kg AA; AD + Pred, positive control mice treated with 5 mg/kg prednisolone in 100 µL of sterilized drinking water.
Page 7:
Figure 4. Effect of asiatic acid (AA) on immune organs in the atopic dermatitis (AD) animal model. (A) The lymph nodes and spleens of the AD mice were photographed on day 18 after treatment. (B) Descriptive statistics showing differences in body, lymph node, and spleen weights in each group. The significant values according to Duncan’s multiple range test (p < 0.05) are indicated by different letters, and values are expressed as the mean ± standard deviation (n = 7 per group). CON, control group; AD only, mice sensitized with 0.25% 2,4-dinitrochlorobenzene (DNCB) in 200 µL of 3:1 acetone/bean oil; AD + AA-L and AD + AA-H, 30 and 75 mg/kg AA-treated DNCB-sensitized mice, respectively, in 100 µL of sterilized drinking water; AA-H, mice orally administered 75 mg/kg AA; AD + Pred, positive control mice treated with 5 mg/kg prednisolone in 100 µL of sterilized drinking water.
Page 8:
Figure 5. Effects of asiatic acid (AA) on dermal and epidermal thickness, collagen fibers, and mast cell infiltration were assessed histologically. (A) The atopic dermatitis (AD) skin lesions were stained using hematoxylin and eosin (H&E), Masson’s trichrome, or toluidine blue, and were photographed under a light microscope at 100× magnification. (B, C, D, E) The dermal and epidermal thicknesses were evaluated using H&E-stained dorsal and ear tissue microphotographs. (F) Collagen fibers were evaluated using Masson’s trichrome staining. (G) Mast cells were counted after toluidine blue staining; the blue points denote mast cells. Values were significant according to Duncan's multiple range test (p < 0.05), and the values are represented as the mean ± standard deviation (n = 7 per group). CON, control group; AD only, mice sensitized with 0.25% 2,4-dinitrochlorobenzene (DNCB) in 200 µL of 3:1 acetone/bean oil; AD + AA-L and AD + AA-H, 30 and 75 mg/kg AA-treated DNCB-sensitized mice, respectively, in 100 µL of sterilized drinking water; AA-H, mice orally administered 75 mg/kg AA; AD + Pred, positive control mice treated with 5 mg/kg prednisolone in 100 µL of sterilized drinking water.
Page 10:
Figure 6. Asiatic acid (AA) effects on atopic dermatitis (AD)-related cytokine expression level in the dorsal skin of 2,4-dinitrochlorobenzene (DNCB)-induced AD mice. (A) Tumor necrosis factor (TNF)-α; (B) interleukin (IL)-1β; (C) IL-12; (D) IL-4; (E) IL-5; (F) IL-6; (G) IL-10; (H) IL-13; (I) IL-31; (J) IL-17; (K) cyclooxygenase (COX)-2; and (L) chemokine ligand (CXCL)-9. The significant values, as per Duncan’s multiple range test (p < 0.05), are represented by different letters. The results are expressed as the mean ± standard deviation (n = 7 per group).
Page 12:
Figure 7. Effects of asiatic acid (AA) on the expression of inflammatory and signaling proteins in the dorsal skin. Expression of (A) anti-inflammatory pathway proteins and (B) nuclear factor kappa B (NF-κB), phosphor(p)-protein kinase B (Akt), p-p38, p-c-Jun N-terminal kinase (JNK), and p-Extracellular-Regulated Protein Kinase 1/2 (ERK1/2) proteins. Different letters indicate significant values per Duncan's multiple range test (p < 0.05), and values are expressed as mean ± standard deviation (n = 7 per group). CON, control group; AD only, mice sensitized with 0.25% 2,4-dinitrochlorobenzene (DNCB) in 200 µL of 3:1 acetone/bean oil; AD + AA-L and AD + AA-H, 30 and 75 mg/kg AA-treated DNCB-sensitized mice, respectively, in 100 µL of sterilized drinking water; AD + Pred, positive control mice treated with 5 mg/kg prednisolone in 100 µL of sterilized drinking water.
- Why don’t you add “*” to label the statically significant values (p < 0.05)
- All statistical analyses were performed using SPSS version 18.0 (IBM; Chicago, IL, USA). Comparisons between experimental groups were performed using one-way analysis of variance with Duncan’s post hoc tests. The result comparisons are represented with a different letter. Also, this statistical method is generally applied in experimentations and manuscript.
Reference paper:
Lee, Y.; Choi, H.K.; N'deh K.P.U.; Choi, Y.J.; Fan, M.; Kim, E.K..; Chung, K.H.; An, J.H. Inhibitory effect of Centella asiatica extract on DNCB-induced atopic dermatitis in HaCaT cells and BALB/c mice. Nutrients. 2020, 12, 411. doi: 10.3390/nu12020411.
Choi H.K.; Kim G.J.; Yoo H.S.; Song D.H.; Chung K.H.; Lee K.J.; Koo Y.T.; An J.H. Vitamin C activates osteoblastgenesis and inhibits osteoclastogenesis via Wnt/β-Catenin/ATF4 signaling pathways. Nutrients. 2019, 11, 506. doi:10.3390/nu11030506
Yoo, H.S.; Kim G.J.; Song D.H.; Chung K.H.; Lee, K.J.; Kim, D.H.; An, J.H. Calcium supplement derived from Gallus gallus domesticus promotes BMP-2/RUNX2/SMAD5 and suppresses TRAP/RANK expression through MAPK signaling activation. Nutrients. 2017, 9, 504. doi:10.3390/nu9050504
Kim, G.J.; Song, D.H.; Yoo H.S.; Chung, K.H.; Lee K.J.; An, J.H. Hederagenin supplementation alleviates the pro-inflammatory and apoptotic response to alcohol in rats. Nutrients. 2017, 9, 41. doi:10.3390/nu9010041
- Please add detail information for each figure. Such as in figure 2C, in the top of each bar was labeled by different letter?
- We added statistic data to the cell viability result in Figure 2A.
Page 5:
- A very small correction, by adding the space between symbols, will need to fix throughout the whole manuscript, e.g. n = 7 per group
- As pointed out in #3, we were written with n = 7 per group while adding information about n.
- Some of the English is rather awkward in places. Hereby, I suggest the author should read the manuscript again carefully.
- This English of manuscript was supplement by native editor with Editage.
- How do the author choose the lower and higher dose (30 and 75 mg/kg/d) and the positive control using for 5 mg/kg prednisolone?
- Generally, there are many types of skin-disease medication including glucocorticoid, Fexofenadine (anti-histamine), and prednisolone. The concern by choosing prednisolone is this medication commonly used to treat allergy and infection. Since our experimentation relates to allergy and infection on the skin, we selected the prednisolone as our positive control. Also, the efficacy of 5 mg/kg prednisolone was demonstrated in the animal model of palladium allergy according to Matsubara R et al (2017), so we used 5 mg/kg prednisolone as a positive control. According to the article entitled “Fexofenadine suppresses delayed-type hypersensitivity in the murine model of palladium allergy”. In the article, the authors used 5mg/kg prednisolone regarding.
- According to our cell viability result, the range concentration from 10 to 20 ㎍/mL has proven to have no cytotoxicity. We calculated the same concentration range of cell viability (10-70 ㎍/mL) to 10-70 mg/kg for the animal experimentation. By this calculation, we were able to choose the low and high dose. We decided to choose 30 mg/kg (low dose) and 75 mg/kg (high dose). Our chosen low and high dose were also supported by article entitled “Effect of route of administration and distribution on drug action,” and “Asiatic acid, a pentacyclic triterpene from Centella asiatica, is neuroprotective in a mouse model of focal cerebral ischemia”. Krishnamurthy, R.G et al (2009) study used 30 and 75 mg/kg in a mouse model as well.
- Reference paper:
Matsubara, R.; Kumagai, K.; Shigematsu, H.; Kitaura, K.; Nakason, Y.; Suzuki, S.; Hamada, Y.; Suzuki R. Fexofenadine suppresses delayed-type hypersensitivity in the murine model of palladium allergy. Molecular sciences. 2017, 18, 1357.
Benet L.Z. Effect of route of administration and distribution on drug action. J Pharmacokinet Biopharm. 1978, 6, 559-585.
Krishnamurthy, R.G.; Senut, M.C.; Zemke D.; Min, J.; Frenkel, M.B.; Greenberg, E.J.; Yu, S.W.; Ahn, N.; Goudreau, J.; Kassab, M. et al. Asiatic acid, a pentacyclic triterpene from Centella asiatica, is neuroprotective in a mouse model of focal cerebral ischemia. J Neurosci Res. 2009, 87, 2541-2550.
- I also encourage the authors to address the comments below. Please change the reference 8 to some chemistry references.
- Thank you for your valuable comments. We have revised the reference section below.
Page 15:
Meeran, M.F.N.; Goyal, S.N.; Suchal, K.; Sharma, C.; Patil, C.R.; Ojha, S.K. Pharmacological properties, molecular mechanisms, and phar-maceutical development of asiatic acid: A pentacyclic triterpenoid of therapeutic promise. Front. Pharmacol. 2018, 9, 892. doi:10.3389/fphar.2018.00892.
- Acebey-Castellon, IL.; Voutquenne-Nazabadioko, L.; Doan Thi Mai, H.; Roseau, N.; Bouthagane, N.; Muhammad, D.; Le Magrex Deba,r, E.; Gamgloff, SC.; Litaudon, M.; Sevenet, T.; et al. Triterpenoid saponins from Symplocos lancifolia. Journal of natural products. 2011. 74, 163-168. doi: 10.1021/np100502y.
- Pakdeechote, P.; Bunbupha, S.; Kukongviriyapan, U.; Prachaney, P.; Khrisanapant, W.; Kukongviriyapan, V. Asiatic acid alleviates hemody-namic and metabolic alterations via restoring eNOS/iNOS expression, oxidative stress, and inflammation in diet-induced metabolic syndrome rats. Nutrients. 2014, 6, 355–370. doi:10.3390/nu6010355.
- Gou, X.J.; Bai, H.H.; Liu, L.W.; Chen, H.Y.; Shi, Q.; Chang, L.S.; Ding, M.M.; Shi, Q.; Zhou, M.X.; Chen, W.L.; et al. Asiatic Acid interferes with invasion and proliferation of breast cancer cells by inhibiting WAVE3 activation through PI3K/AKT signaling pathway. Biomed Res. Int. 2020, 2020, 1874387. doi:10.1155/2020/1874387.
- Huang, S.S.; Chiu, C.S.; Chen, H.J.; Hou, W.C.; Sheu, M.J.; Lin, Y.C.; Shie, P.H. Antinociceptive activities and the mechanisms of an-ti-inflammation of asiatic acid in mice. Evid-based Complement Alternat Med. 2011, 2011, 895857. doi:10.1155/2011/895857.
- Li, Y.; Yang, F.; Yuan, M.; Jiang, L.; Yuan, L.; Zhang, X.; Li, Y.; Dong, L.; Bao, X.; Yin, S. Synthesis and evaluation of asiatic acid derivatives as anti-fibrotic agents: Structure/activity studies. Steroids. 2015, 96, 44–49. doi:10.1016/j.steroids.2014.11.001.
- Hao, C.; Wu, B.; Hou, Z.; Xie, Q.; Liao, T.; Wang, T.; Ma, D. Asiatic acid inhibits LPS-induced inflammatory response in human gingival fibroblasts. Int. Immunopharmacol. 2017, 50, 313–318. doi:10.1016/j.intimp.2017.07.005.
- Lee, Y.; Choi, H.K.; N'deh K.P.U.; Choi, Y.J.; Fan, M.; Kim, E.K..; Chung, K.H.; An, J.H. Inhibitory effect of Centella asiatica extract on DNCB-induced atopic dermatitis in HaCaT cells and BALB/c mice. Nutrients. 2020, 12, 411. doi: 10.3390/nu12020411.
- Hwang, Y.S.; Chang, B.Y.; Kim, T.Y.; Kim, S.Y. Ameliorative effects of green tea extract from tannase digests on house dust mite anti-gen-induced atopic dermatitis-like lesions in NC/Nga mice. Arch. Dermatol. Res. 2019, 311, 109–120. doi:10.1007/s00403-018-01886-6.
- Choi, E.J.; Iwasa, M.; Han, K. I; Kim, W.J.; Tang, Y.; Hwang, Y.J.; Chae, J.R.; Han, W.C.; Shin, Y.S.; Kim, E.K. Heat-killed enterococcus faecalis EF-2001 ameliorates atopic dermatitis in a murine model. Nutrients. 2016, 8, 146. doi:10.3390/nu8030146.
- Boniface, K.; Bernard, F.X.; Garcia, M.; Gurney, A.L.; Lecron, J.C.; Morel, F. IL-22 inhibits epidermal differentiation and induces proin-flammatory gene expression and migration of human keratinocytes. J. Immunol. 2005, 174, 3695–3702. doi:10.4049/jimmunol.174.6.3695.
- Yang, H.R.; Lee, H.; Kim, J.H.; Hong, I.H.; Hwang, D.H.; Rho, I.R.; Kim, G.S.; Kim, E.; Kang, C. Therapeutic effect of Rumex japonicus Houtt. on DNCB-induced atopic dermatitis-like skin lesions in Balb/c mice and human keratinocyte HaCaT cells. Nutrients. 2019, 11, 573. doi:10.3390/nu11030573.
- De Vuyst, E.; Salmon, M.; Evrard, C.; Lambert de Rouvroit, C.; Poumay, Y. Atopic Dermatitis studies through in vitro models. Front. Med. 2017, 4, 119. doi:10.3389/fmed.2017.00119.
- Galli, S.J.; Tsai, M. IgE and mast cells in allergic disease. Nat. Med. 2012, 18, 693–704. doi:10.1038/nm.2755.
- Lee, J.H.; Jeon, Y.D.; Lee, Y.M.; Kim, D.K. The suppressive effect of puerarin on atopic dermatitis-like skin lesions through regulation of inflammatory mediators in vitro and in vivo. Biochem Biophys Res Commun. 2018, 498, 707–714. doi:10.1016/j.bbrc.2018.03.018.
- Mechesso, A.F.; Lee, S.J.; Park, N.H.; Kim, J.Y.; Im, Z.E.; Suh, J.W.; Park, S.C. Preventive effects of a novel herbal mixture on atopic der-matitis-like skin lesions in BALB/C mice. BMC Complement. Altern. Med. 2019, 19, 1–13. doi:10.1186/s12906-018-2426-z.
- Liu, F.T.; Goodarzi, H.; Chen, H.Y. IgE, mast cells, and eosinophils in atopic dermatitis. Clin Rev Allergy Immunol. 2011, 41, 298–310. doi:10.1007/s12016-011-8252-4.
- Nakae, S.; Suto, H.; Iikura, M.; Kakurai, M.; Sedgwick, J.D.; Tsai, M.; Galli, S.J. Mast cells enhance T cell activation: importance of mast cell costimulatory molecules and secreted TNF. J. Immunol. 2006, 176, 2238–2248. doi:10.4049/jimmunol.176.4.2238.
- Shea-Donohue, T.; Stiltz, J.; Zhao, A.; Notari, L. Mast cells. Curr Gastroenterol Rep. 2010, 12, 349–357. doi:10.1007/s11894-010-0132-1.
- Piera, L.; Olczak, S.; Kun, T.; Galdyszynska, M.; Ciosek, J.; Szymanski, J.; Drobnik, J. Disruption of histamine/H3 receptor signal reduces collagen deposition in cultures scar myofibroblasts. J Physiol Pharmacol. 2019, 70, 239–247. doi:10.26402/jpp.2019.2.07.
- Hong, S.; Lee, B.; Kim, J.H.; Kim, E.Y.; Kim, M.; Kwon, B.; Cho, H.R.; Sohn, Y.; Jung, H.S. Solanum nigrum Linne improves DNCB-induced atopic dermatitis-like skin disease in BALB/c mice. Mol. Med. Rep. 2020, 22, 2878–2886. doi:10.3892/mmr.2020.11381.
- Puniya, B.L.; Todd, R.G.; Mohammed, A.; Brown, D.M.; Barberis, M.; Helikar, T. A mechanistic computational model reveals that plasticity of CD4+ T cell differentiation is a function of cytokine composition and dosage. Front. Physiol. 2018, 9, 878. doi:10.3389/fphys.2018.00878.
- Hamid, Q.; Boguniewicz, M.; Leung, D.Y. Differential in situ cytokine gene expression in acute versus chronic atopic dermatitis. J.Clin. Invest. 1994, 94, 870–876. doi: 10.1172/JCI117408.
- Szegedi, A.; Baráth, S.; Nagy, G.; Szodoray, P.; Gál, M.; Sipka, S.; Bagdi, E.; Banham, A.H.; Krenács, L. Regulatory T cells in atopic dermatitis: epidermal dendritic cell clusters may contribute to their local expansion. Br. J. Dermatol. 2009, 160, 984–993. doi:10.1111/j.1365-2133.2009.09035.x.
- Huang, W.C.; Huang, C.H.; Hu, S.; Peng, H.L.; Wu, S.J. Topical spilanthol inhibits MAPK signaling and ameliorates allergic inflammation in DNCB-induced atopic dermatitis in mice. Int. J. Mol. Sci. 2019, 20, 2490. doi:10.3390/ijms20102490.
- Bieber, T. Interleukin-13: Targeting an underestimated cytokine in atopic dermatitis. Allergy. 2020, 75, 54–62. doi:10.1111/all.13954.
- Gil, T.Y.; Kang, Y.M.; Eom, Y.J.; Hong, C.H.; An, H.J. Anti-atopic dermatitis effect of Seaweed Fulvescens extract via inhibiting the STAT1 pathway. Mediators Inflamm. 2019, 2019, 3760934. doi:10.1155/2019/3760934.
- Lee, Y.J.; Oh, M.J.; Lee, D.H.; Lee, Y.S.; Lee, J.; Kim, D.H.; Choi, C.H.; Song, M.J.; Song, H.S.; Hong, J.T. Anti-inflammatory effect of bee venom in phthalic anhydride-induced atopic dermatitis animal model. Inflammopharmacology. 2020, 28, 253–263. doi:10.1007/s10787-019-00646-w.
- Herlaar, E.; Brown, Z. p38 MAPK signalling cascades in inflammatory disease. Mol. Med. Today. 1999, 5, 439–447. doi:10.1016/S1357-4310(99)01544-0.
- Yi, C.; Si, L.; Xu, J.; Yang, J.; Wang, Q.; Wang, X. Effect and mechanism of asiatic acid on autophagy in myocardial ischemia‑reperfusion injury. Exp. Ther. Med. 2020, 20, 54. doi:10.3892/etm.2020.9182.
- Johansen, C.; Kragballe, K.; Westergaard, M.; Henningsen, J.; Kristiansen, K.; Iversen, L. The mitogen-activated protein kinases p38 and ERK1/2 are increased in lesional psoriatic skin. Br. J. Dermatol. 2005, 152, 37–42. doi:10.1111/j.1365-2133.2004.06304.x.
- Qin, W.; Duan, J.; Xie, X.; Kang, J.; Deng, T.; Chen, M. Exposure to diisononyl phthalate promotes atopic march by activating of NF-κB and p38 MAPK. Toxicol. Appl. Pharmacol. 2020, 395, 114981. doi:10.1016/j.taap.2020.114981.
- Ye, J.; Piao, H.; Jiang, J.; Jin, G.; Zheng, M.; Yang, J.; Jin, X.; Sun, T.; Choi, Y.H.; Li, L.; et al. Polydatin inhibits mast cell-mediated allergic inflammation by targeting PI3K/Akt, MAPK, NF-κB and Nrf2/HO-1 pathways. Sci. Rep. 2017, 7, 1–13. doi:10.1038/s41598-017-12252-3.
- Chen, L.; Chen, J.; Xie, C.M.; Zhao, Y.; Wang, X.; Zhang, Y.H. Maternal Disononyl Phthalate Exposure Activates Allergic Airway Inflam-mation via Stimulating the Phosphoinositide 3-kinase/Akt Pathway in Rat Pups. Biomed. Environ. Sci. 2015, 28, 190–198. doi:10.3967/bes2015.025.
- Yun, K.J.; Kim, J.Y.; Kim, J. B; Lee, K.W.; Jeong, S.Y.; Park, H.J.; Jung, H.J.; Cho, Y.W.; Yun, K.; Lee, K.T. Inhibition of LPS-induced NO and PGE2 production by asiatic acid via NF-κB inactivation in RAW 264.7 macrophages: possible involvement of the IKK and MAPK pathways. Int. Immunopharmacol. 2008, 8, 431–441. doi:10.1016/j.intimp.2007.11.003.
- P2L55 the number 4 of the word CCl4 should be lowercase.
- In this manuscript, CC14 is only specified the introduction part, so we have been revised CC14 abbreviations to carbon tetrachloride.
Page 2:
In a carbon tetrachloride-induced liver damage rat model, AA was shown to have an anti-fibrotic effect as it reduces aspartate aminotransferase activity and alanine ami-notransferase activity in the serum [13]
- In figure 5A, would it be possible to use different arrow to point out the epidermal and dermal thickness to show the difference?
- In Figure 5A, the thickness of epidermal and epidermal are expressed by green arrow (epidermal layer) and black-bolded arrow (dermal layer).
Page 8:
- Why don’t the author test the protein expression of ERK1/2 and JNK?
- We have supplemented our revised ERK1/2 and JNK data along with the descriptions as well.
Page 1:
Abstract: We examined the immunomodulatory and anti-inflammatory effects of asiatic acid (AA) in atopic dermatitis (AD). AA treatment (5–20 µg/mL) dose-dependently suppressed the tumor necrosis factor (TNF)-α level and interleukin (IL)-6 protein expression in interferon (IFN)-γ + TNF-α-treated HaCaT cells. The 2,4-dinitrocholrlbenzene (DNCB)-induced AD animal model was developed by administering two AA concentrations (30 and 75 mg/kg/d: AD + AA-L and AD + AA-H groups, respectively) for 18 days. Interestingly, AA treatment decreased AD skin lesions formation and affected other AD characteristics, such as increased ear thickness, lymph node and spleen size, dermal and epidermal thickness, collagen deposition, and mast cell infiltration in dorsal skin. In addition, in the DNCB-induced AD animal model, AA treatment downregulated the mRNA expression level of AD-related cytokines, such as Th1- (TNF-α and IL-1β and -12) and Th2 (IL-4, -5, -6, -13, and -31)-related cytokines as well as that of cyclooxygenase-2 and CXCL9. Moreover, in the AA treatment group, the protein level of inflammatory cytokines, including COX -2, IL-6, TNF-α, and IL-8, as well as the NF-κB and MAPK signaling pathways, were decreased. Overall, our study confirmed that AA administration inhibited AD skin lesion formation via enhancing immunomodulation and inhibiting inflammation. Thus, AA can be used as palliative medication for regulating AD symptoms.
Page 4:
2.5. Western Blotting
The cells and mouse dorsal tissues were homogenized using lysine buffers containing protease inhibitors (Roche; Mannheim, Germany). The total soluble protein content was assessed using the Bio-Rad protein kit (Bio-Rad Laboratories; Hercules, CA, USA). The proteins were electrophoresed and transferred to Immobilon-P transfer membranes (Millipore; Burlington, MA, USA). The membranes were blocked with 5% bovine serum albumin and incubated at 4°C for 24 h with specific primary antibodies against phosphorylated-p38 (p-p38), p-protein kinase B (Akt), p-n N-terminal kinase (JNK), p-extracellular signal-regulated kinase 1/2 (ERK1/2), β-actin (Cell Signaling Technology; Beverly, MA, USA), TNF-α, NF-kB, iNOS, COX-2 (Abcam, Cambridge, MA, USA), IL-6, and IL-8 (Santa Cruz Biotechnology; Santa Cruz, CA, USA). The membranes were then incubated at 4°C with goat anti-rabbit IgG (H + L) and a horseradish peroxidase-conjugated secondary antibody (Abcam). The protein bands were visualized using enhanced chemiluminescence, and densitometric analysis of the protein bands was performed using the C-DiGit Blot Scanner (Li-COR; Lincoln, NE, USA) and ImageJ software (NIH; Rockville, MD, USA). All data were normalized to the β-actin values.
Page 12:
Figure 7. Effects of asiatic acid (AA) on the expression of inflammatory and signaling proteins in the dorsal skin. Expression of (A) anti-inflammatory pathway proteins and (B) nuclear factor kappa B (NF-κB), phosphor(p)-protein kinase B (Akt), p-p38, p-c-Jun N-terminal kinase (JNK), and p-Extracellular-Regulated Protein Kinase 1/2 (ERK1/2) proteins. Different letters indicate significant values per Duncan's multiple range test (p < 0.05), and values are expressed as mean ± standard deviation (n = 7 per group). CON, control group; AD only, mice sensitized with 0.25% 2,4-dinitrochlorobenzene (DNCB) in 200 µL of 3:1 acetone/bean oil; AD + AA-L and AD + AA-H, 30 and 75 mg/kg AA-treated DNCB-sensitized mice, respectively, in 100 µL of sterilized drinking water; AD + Pred, positive control mice treated with 5 mg/kg prednisolone in 100 µL of sterilized drinking water.
Page 12:
3.7. Effects of AA Treatment on NF‐κB, p-Akt, and MAPK Signaling in an AD Animal Model
The effects of AA treatment on the expression of NF‐κB, p-Akt, and MAPK signaling in the dorsal tissues of AD mice were also determined (Figure 7B). The inhibition of NF-κB and MAPKs is critically related to the decrease of the anti-inflammatory response and mast cell count [23]. The NF-κB protein expression level in the AD only group was increased (50%) compared with that of the CON group. AA treatment dose-dependently reduced NF-κB protein expression (10.8% and 56.5%, respectively, in AD + AA-L and AD + AA-H groups) compared with that in the AD only group. Furthermore, the NF-κB level in the AD + AA-H group was decreased by 6.6%, which was higher than that in the CON group. Interestingly, the NF-κB level in the AD + AA-H group (33%) was decreased compared with that in the AD + Pred group. The expression level of p-Akt protein in the AD only group was increased by 44%, which was higher than that in the CON group. In the AA-treated groups (AD + AA-L and AD + AA-H groups), the p-Akt expression was decreased by 25% and 51%, respectively, compared with that in the AD only group. Interestingly, the p-Akt expression level in the AD + AA-H group (7%) was downregulated compared with that in the CON group. The MAPK (p-p38, p-JNK, and p-ERK1/2) protein expression in the AD only group was 40%, 24%, and 23% higher, respectively, compared with that in the CON group. Moreover, the AD + AA-L and AD + AA-H group of the p-p38 (25% and 52%, respectively), p-JNK (35% and 90%, respectively), and p-ERK1/2 (38% and 33%, respectively) levels were decreased compared with that in the AD only group. NF-κB expression and MAPK signaling were dose-dependently decreased after AA treatment. Our results demonstrate that AA treatment inhibited allergic inflammation by restraining NF‐κB, p-Akt, and MAPK signaling levels in the AD model.
Page 13:
In this study, we evaluated whether AA treatment downregulates Th1- and Th2-related cytokine levels and reduces NF-κB, p-Akt, and MAPK signaling 1evels in a DNCB-induced AD animal model (Figure 6 and 7).
Page 14:
In LPS-induced macrophages, AA inhibits IL-6, -1β, and TNF-α expression via the p38, ERK1/2, JNK, and NF-κB pathways [42]. This study showed that AA reduced the expression of NF-κB, p-Akt, and MAPK signaling pathways in DNCB-induced AD mice (Figure 7). Thus, our results suggest that AA treatment reduces inflammation related gene by downregulating the NF-κB and MAPK signaling pathways.
- Conclusions
This is the first study to investigate the immunomodulatory and anti-inflammatory effects of AA in an AD animal model. We suggest that 20 µg/mL AA downregulates in-flammatory cytokines in TNF-α+IFN-γ-treated HaCaT cells. AA treatment for 18 days reg-ulated swelling, excoriation, edema, scarring in both ears and dorsal skin in DNCB-induced AD mice. AA administration also reduced lymph and spleen weight, dermal and epidermal thickness, collagen deposition, and mast cell count in a DNCB-induced AD animal model, thereby reducing skin lesions due to allergic reactions. Furthermore, AA was proved to have immunomodulatory and anti-inflammatory effects as it blocked Th1- and Th2-related cytokines. In addition, inflammatory cytokines as well as the NF-κB, p-Akt, and MAPK signaling pathways were suppressed in the AD+AA-H group. In conclusion, our results suggested that AA administration reduces AD skin le-sions by enhancing immunomodulation and inhibiting inflammation and might be a po-tent therapy for attenuating AD symptoms such as itching, swelling, crust formation, and leathery skin. This is the first study to investigate the immunomodulatory and anti-inflammatory effects of AA in an AD animal model. We suggest that 20 µg/mL AA downregulates inflammatory cytokines in TNF-α + IFN-γ-treated HaCaT cells. AA treatment for 18 days regulated swelling, excoriation, edema, and scarring in both ears and dorsal skin in DNCB-induced AD mice. AA administration also reduced lymph and spleen weight, dermal and epidermal thickness, collagen deposition, and mast cell count in a DNCB-induced AD animal model, thereby reducing the formation of skin lesions due to allergic reactions. Furthermore, AA was proved to have immunomodulatory and anti-inflammatory effects as it blocked the activity of Th1- and Th2-related cytokines. In addition, the levels of inflammatory cytokines as well as the activity of NF-κB, p-Akt, and MAPK signaling pathways were suppressed in the AD + AA-H group. In conclusion, our results suggest that AA administration reduces the formation of AD skin lesions by enhancing immunomodulation and inhibiting inflammation and might be a potent therapy for attenuating AD symptoms such as itching, swelling, crust formation, and leathery skin.

Round 2
Reviewer 2 Report
The authors had revised all the comments and also provide more clear figures. However, the author need make a small correction:
In the abstract, please add COX-2 after the word cyclooxygenase-2